# Towards Mass-Scale IoT with Energy-Autonomous LoRaWAN Sensor Nodes

**DOI:** 10.3390/s24134279

**Published:** 2024-07-01

**Authors:** Roberto La Rosa, Lokman Boulebnane, Antonino Pagano, Fabrizio Giuliano, Daniele Croce

**Affiliations:** 1STMicroelectronics, Stradale Primosole 50, 95121 Catania, Italy; 2Department of Engineering, University of Palermo, 90128 Palermo, Italy; lokman.boulebnane@community.unipa.it (L.B.); fabrizio.giuliano@unipa.it (F.G.); daniele.croce@unipa.it (D.C.); 3Palermo Research Unit, CNIT (National Inter-University Consortium for Telecommunications), 90128 Palermo, Italy

**Keywords:** battery-free, energy harvesting, IoT, wireless sensor node, LPWAN, LoRaWAN

## Abstract

By 2030, it is expected that a trillion things will be connected. In such a scenario, the power required for the trillion nodes would necessitate using trillions of batteries, resulting in maintenance challenges and significant management costs. The objective of this research is to contribute to sustainable wireless sensor nodes through the introduction of an energy-autonomous wireless sensor node (EAWSN) designed to be an energy-autonomous, self-sufficient, and maintenance-free device, to be suitable for long-term mass-scale internet of things (IoT) applications in remote and inaccessible environments. The EAWSN utilizes Low-Power Wide Area Networks (LPWANs) via LoRaWAN connectivity, and it is powered by a commercial photovoltaic cell, which can also harvest ambient light in an indoor environment. Storage components include a capacitor of 2 mF, which allows EAWSN to successfully transmit 30-byte data packets up to 560 m, thanks to opportunistic LoRaWAN data rate selection that enables a significant trade-off between energy consumption and network coverage. The reliability of the designed platform is demonstrated through validation in an urban environment, showing exceptional performance over remarkable distances.

## 1. Introduction

Wireless sensor nodes (WSNs) are rapidly becoming the most critical elements in IoT infrastructure for several domain-based solutions, aiming to enhance the quality of life by integrating cutting-edge technologies. Wireless devices possess versatile computing, communication, and control capabilities [1], making them adaptable to a wide range of IoT applications. While the requirements for WSNs can vary significantly, defining fixed criteria for nodes and networks is impractical. Nonetheless, common characteristics define WSNs, including deployment adaptability, mobility support, compactness, lightweight design, energy efficiency, wireless communication capability, and power optimization [2]. As the demands on wireless nodes continue to grow, these include increasingly performing complex and advanced functions [3]. Battery-powered sensor devices are commonly used in WSNs and often present challenges of power consumption and the need to replace batteries [4], which is a critical issue. Ongoing research efforts aim at increasing the energy density of batteries, and the search is on for solutions to extend the operational life of sensor nodes. However, even with the most advanced technology available, the lifetime of the entire network remains limited, typically ranging from a few months to a few years [5].

A promising approach to achieving perpetual and sustainable network operation is using energy-autonomous, battery-less wireless systems powered solely by energy harvesting (EH) from environmental sources like light, radio waves, temperature variations, vibrations, motion, wind, and water currents. These approaches provide a practical solution for efficiently harnessing energy from the environment, ensuring the sustainable power supply for wireless sensor nodes, and enabling uninterrupted network operation [1,4,6,7]. Energy-autonomous wireless systems focus on efficiency through the use of a variety of ultra-low-power design techniques. These techniques include the use of low-power microcontrollers and the implementation of operating modes that minimize or eliminate power consumption or efficient power control mechanisms to optimize internal energy use [3,6,8,9].

In WSNs, communication is generally the most energy-consuming operation, mainly due to the high power consumption associated with radio transmissions [5]. Adaptive transmission algorithms play a vital role in optimizing the efficiency and dependability of these sensors to address this issue. The goal is to find a trade-off between energy consumption and communication demands by considering design parameters such as transmission power, data rate, modulation scheme, and duty cycle [6]. A current trend is to adopt energy-efficient communication technologies such as ZigBee, Bluetooth Low Energy (BLE), and LoRaWAN. In particular, LoRaWAN is used in various applications due to its exceptional range and energy efficiency. It operates in the license-free Industrial, Scientific, and Medical (ISM) band [10,11,12]. LoRaWAN, on the other hand, builds on LoRa to provide long-range communications between LoRa devices and network gateways [13,14]. The literature presents numerous energy-conscious transmission algorithms designed to enhance the performance of battery-free IoT sensors [15,16,17,18]. Adaptive transmission algorithms are pivotal in this context, as they play a crucial role in maximizing both efficiency and reliability [6,9]. These approaches take into account parameters such as transmission power, data rate, and the modulation scheme (e.g., spreading factor and bandwidth) while using off-the-shelf components to eliminate the need for battery replacement in devices located in hazardous areas [3,19].

Taking advantage of the widespread availability and cleanliness of this renewable resource, this paper presents a novel battery-free EAWSN platform powered by solar energy. The platform comprises an STM32WL Nucleo 64 board with an STM32WLJC1 MCU that can support LPWAN with a dual-core 32-bit architecture (ARM CortexM4/M0) and a radio frequency (RF) transceiver that operates from 150 MHz through 960 MHz. Our design also incorporates a 2 mF to 16 mF amorphous silicon solar cell (AM1815) and storage condenser ($C_{storage}$) for testing and validation. We argue in favor of a measurement-based approach, which we consider to be more practical than mathematical models or simulation-based studies. In addition, our work is original in optimizing network parameters such as packet size and dedicated storage allocation along with the LoRaWAN configuration, tailoring each setup for self-powered and battery-free LoRaWAN devices. Finally, our work aims to address the following research questions: (I) How does a battery-free LoRaWAN sensor perform in terms of coverage and signal quality in an urban environment? (II) How does energy storage capacity influence the packet length and spreading factor of the battery-free sensor? (III) What are the limitations of real-world deployment of battery-free sensors in urban environments? (IV) How can these limitations be overcome through optimization of system design?

Our paper is structured in the following way: Section 2 presents the related works, focusing on the LoRaWAN background and battery-free devices for IoT. In Section 3, we analyze the architectural design of our EAWSN system, explaining its main components description, design, and detailed description of the microcontroller. Section 4 outlines the experimental objectives and setup, providing detailed information on tools, equipment, and base station (BS) configuration to effectively exploit the LoRaWAN protocol through a gateway and network server. Section 5 focuses on analyzing the experimental data, showing how variations in LoRaWAN parameters affect energy consumption, range, and packet length (PL). The results show the performance and efficiency of the system in transmitting data over a range of distances, in both base station and gateway environments, and within the power constraints imposed by the battery-free design. Section 6 discusses the recommended settings for the EAWSN under different lighting conditions, addresses possible limitations, and examines its mass scale capability. Finally, the concluding section summarizes the main findings and contributions.

## 2. Related Works

This section presents the background on LoRaWAN and provides a brief literature review on energy-autonomous and battery-free sensors for IoT, highlighting the potentialities and the challenges of this technology.

### 2.1. Background on LoRaWAN

LoRa technology represents a significant evolution in wireless communications, specifically designed to meet the needs of the IoT [20]. It is characterized by the ability to transmit data over long distances while using minimal energy [13]. LoRa is based on spread spectrum modulation technology, known for its remarkable signal propagation capabilities, ensuring reliable communications in diverse environments, including urban, rural, and industrial landscapes [21,22]. By optimizing the delicate balance between communication range and power efficiency, LoRa demonstrates adaptability through its control over transmit power, modulation rate, and payload size, making it highly suitable for various application scenarios [23]. Furthermore, the unique architecture of LoRa enables the creation of extensive networks over long distances, making it an effective choice for a variety of applications, such as smart cities, agriculture, asset tracking, and environmental monitoring [24]. LoRaWAN comprises two essential layers: the physical layer, utilizing Semtech’s patented Chirp Spread Spectrum modulation (CSS), and the Media Access Control (MAC) layer, governed by the LoRa Alliance-defined LoRaWAN protocol. These combined technical aspects establish LoRa as a disruptive force within the realms of IoT and long-range wireless communications [25]. One of the most adopted MAC layers within LoRa technology is LoRaWAN [26,27]. It plays a pivotal role in overseeing access to shared media and ensuring efficient, dependable data transmission among connected devices [14]. This standard is designed to enhance communication across wide-area networks. As depicted in Figure 1a, the architecture of the LoRaWAN network shows how end devices establish communication with the gateway through the LoRa RF interface. Subsequently, the gateway transmits frames to the server via non-LoRaWAN networks like Ethernet, Cellular (e.g., 3G/4G/5G), Wi-Fi, and similar alternatives [28]. Figure 1b showcases the communication stack specific to LoRaWAN, where the physical layer delineates the ISM band (e.g., 868 MHz in Europe).

The LoRaWAN specification outlines three distinct classes designed to address various application demands. These classes determine the communication protocols between devices and the network and regulate power consumption management. An overview of each LoRaWAN device class can be found in [14] and is summarized in Table 1. Choosing which class to use depends on the specific requirements of the IoT application. Class A is ideal for battery-powered devices with minimal power consumption requirements [29,30], even if this means occasional downlink latency. For applications that require more predictable downlink communication at the expense of higher power consumption, Class B is suitable. Class C is used when low-latency downlink communication is essential and it is possible to relax the power restrictions.

The spreading factor (SF) is a critical parameter in the LoRa physical layer that defines the ratio between symbol rate (Rs) Equation (Equation 1) and chip rate (Rc) Equation (Equation 2),
(1)$$Rs=\frac{1}{Ts}=\frac{BW}{2^{SF}}\;\mathrm{symbols}/\sec$$
(2)$$Rc=Rs\cdot 2^{SF}\;\mathrm{chips}/\sec$$

SF determines the number of chips used to encode a symbol. For each SF, Table 2 illustrates $2^{SF}$ chips per symbol.

Increasing the spreading factor (SF) value results in a higher sensitivity of the receiver and extends the range. However, this adjustment reduces the data bit rate (as illustrated in Equation (Equation 4)) and increases the time on air (ToA) of the packet. CR denotes the code rate determining the extent of forward error correction (FEC). LoRa provides CR values ranging from 0 to 4, with CR = 0 denoting no FEC.
(3)Sensitivity=−174+10log(BW)+SNR+NF
(4)$$Rb=SF\cdot \frac{BW}{2^{SF}}\cdot \frac{4}{4+RC}\;\mathrm{bits}/\sec$$

Each incremental increase in spreading factor (SF) halves the transmission rate, doubling the transmission time and consequently increasing the power consumption [31,32].

LoRaWAN has a set of six orthogonal spreading factors in the range of 7 to 12, providing a dynamic balance between data rate and communication range, as represented in Table 2. Note that LoRA network conditions (e.g., power transmission or end-device position) can have significant impact on orthogonality of SF allocations [33,34,35].

### 2.2. Internet of Battery-Less Things

Batteries have limitations, presenting several challenges. As we progress into a future marked by countless IoT devices, this would lead to an economic burden due to the need to replace large quantities of depleted batteries and devices. Furthermore, it would also pose significant environmental concerns [36]. Indeed, an expansion of IoT based on trillions of new battery-powered devices could lead to an environmental catastrophe. Most of the batteries that are disposed of are deposited in landfills, with a mere 5% undergoing recycling [36,37]. As these discarded batteries break down, they emit harmful fumes into the atmosphere and leach chemicals into the soil. Additionally, the process of battery recycling introduces pollutants into water bodies [37].

Over the last decade, research has focused on a new system, widely deployed and battery-free. Similar to conventional sensors, battery-free IoT devices are equipped with modules for detection, processing, and communication. Instead of relying on batteries storing chemical energy, these devices use small capacitors as energy buffers. For instance, Delgado et al. [38] introduced a Markov model for assessing the performance of battery-free devices in uplink and downlink (UL/DL) transmissions, assessing their effectiveness regarding factors like device setup and environmental circumstances. The study demonstrated that with a 47 mF capacitor and a 1 mW energy harvesting rate, it is feasible to sustain 1 Byte transmissions every 60 s.

Moreover, in [39], the authors investigated the efficacy of energy-harvested battery-free sensors in case of random transmission patterns. Employing stochastic geometry and Markov chain analysis techniques, a mathematical model was devised for each system component, allowing for the analytical determination of the probabilities associated with energy and communication failures. The research underscored that the adaptive data rate (ADR) feature in LoRa networks could result in energy deficits when utilizing higher spreading factors. Consequently, they proposed adaptive charging time strategies as a potential remedy. In their work [40], the authors explored the ideal parameters for scheduling application tasks on battery-less IoT sensor devices. Through an environment emulator, they validated a mathematical model for selecting these parameters, aiming to minimize the application cycle completion time for sensing and transmission tasks under varying device and environmental conditions. Their analysis indicated that a device fitted with a 10 mF capacitor can perform temperature measurements and transmit data at intervals of at least once every 5 s while being capable of harvesting a minimum of 50 mW (equivalent to 10 mA of current). Battery-free IoT devices operate intermittently due to the high variability and unpredictability of the harvested ambient energy, leading to frequent power interruptions. This operating pattern can hinder the normal progression of computational operations, as energy interruptions prevent continuous monitoring. In these cases, the combined use of supercapacitors and adaptive algorithms can optimize monitoring even by dynamically adjusting key network parameters, including packet transmission time, data redundancy, and packet size, to enhance device performance, for instance, optimizing data monitoring and transmission even when the natural energy source is absent, such as during the night [6].

In the study by Sabovic et al. [40], the researchers delved into determining the best parameters for scheduling application tasks on battery-less IoT sensor devices. Utilizing an environment emulator, they confirmed the accuracy of a mathematical model designed to select these optimal parameters, to achieve the shortest completion time for application cycles, thereby facilitating sensing and transmission tasks across diverse device and environmental conditions. Their analysis demonstrated that a device furnished with a 10 mF capacitor is capable of performing temperature measurements and transmitting data at intervals of no more than once every 5 s, while being able to harvest a minimum of 50 mW (equivalent to 10 mA of current). Battery-free IoT devices operate intermittently due to the highly variable and unpredictable nature of harvested ambient energy, resulting in frequent power disruptions. Such operational interruptions can impede the smooth flow of computational tasks, as discontinuous energy availability prevents continuous monitoring. In such scenarios, employing supercapacitors in conjunction with adaptive algorithms can optimize monitoring by dynamically adjusting key network parameters like packet transmission timing, data redundancy, and packet size, thereby enhancing device performance, for instance, optimizing data monitoring and transmission even during periods of natural energy absence, such as nighttime, as discussed in [6]. Finally, to the best of our knowledge, this study represents the first real-world validation of battery-free wireless sensors in a complex urban environment. However, these theoretical results, and in particular [38,39], might be overly optimistic in terms of interarrival times between packets and the size of the storage capacitor, requiring to be confirmed in a real-world implementation.

## 3. Design of the Energy-Autonomous and Battery-Free LoRaWAN Sensor Node

In a standard wireless sensor network (WSN), a sensor without a conventional battery must efficiently harvest, control, and store energy to perform tasks such as sensing, data processing, and wirelessly transmitting data to a remote base station.

The basic architecture of an energy-autonomous wireless node (EAWSN), powered solely by energy harvested from the environment, is shown in the block diagram in Figure 2. This diagram also shows the LORIOT cloud which includes a network server (NS) and an application server (AS). The LORIOT LoRaWAN network server integrates all the main output formats and several IoT platforms.

Figure 3 provides a visual depiction of the operation of the EAWSN, illustrating the alternating phases of energy harvesting and data transmission. In the energy harvesting phase, the node harvests energy from ambient light via the photovoltaic transducer and accumulates it in the storage capacitor ($C_{storage}$), gradually increasing the voltage ($V_{store}$). The energy stored is subsequently utilized to power the LoRaWAN radio during the data transmission phase.

The key design challenge in developing an EAWSN revolves around efficiently managing these limited energy resources, which, in this work, is primarily provided by the light-induced current generated by the photovoltaic cell.

Energy management in an EAWSN is the principal design challenge because the system relies uniquely on energy harvested from the environment, which is inherently limited. Therefore, the power consumption and energy efficiency of each component within the system are critical. For the system to operate sustainably and autonomously, it must ensure efficient energy utilization. For this reason, the design of these systems utilizes ultra-low-power architectures, with particular attention paid to the appropriate selection of devices in terms of power consumption. Based on these principles, the STM32WL5 microcontroller (by STMicroelectronics) was selected, as it is an ultra-low-power microcontroller device that offers comprehensive support for eight primary low-power modes, including Low-Power Run, Sleep, Low-Power Sleep, Stop 0, Stop 1, Stop 2, STANDBY with RAM retention, Standby, and Shutdown. Within each of these main modes, several configurable sub-modes provide a range of power-saving options [41]. The STM32WL5 microcontroller includes a programmable voltage detector (PVD) that can operate while the MCU is in a low-power mode. This PVD is fundamental for the development of battery-free sensors. The PVD continuously monitors the voltage level of the storage voltage ($V_{store}$). When the $V_{store}$ reaches its highest value $V_{H}$, the system switches from the energy harvesting phase to the data transmission phase.

During the data transmission phase, the current required to power the radio is tens of milliamperes, which exceeds, by more than an order of magnitude, the photovoltaic transducer’s current generation, typically in the order of tens of microamperes. As a result, ($V_{store}$) drops rapidly and reaches its lowest point, $V_{L}$. To ensure uninterrupted operation and prevent MCU from resetting, $V_{L}$ must never fall below the minimum MCU bias voltage, $V_{dd\_min}$. Therefore, the design must ensure the following conditions for the voltage $V_{L}$ and the storage capacitor $C_{storage}$, as expressed by Equations (Equation 5) and (Equation 6).
(5)$$V_{L}>V_{dd\_min}$$
(6)$$C_{storage}>\frac{1}{2}\cdot E_{LoRa\_TX}\cdot \left(V_{H}^{2}-V_{L}^{2}\right)$$
where the energy required to transmit a data packet, denoted as $E_{LoRa_{T}X}$, is subject to various influencing factors. Among these, critical factors affecting energy consumption for transmitting LoRaWAN data packets include packet length, spreading factor (SF), bandwidth (BW), coding rate, and transmitted power. During the energy harvesting phase, the total quiescent current ($I_{q}$) of the system is approximately 1 μA, a crucial performance threshold for the limit of detection (LoD) of the system. While actively harvesting energy, the $C_{storage}$ capacitor is charged by the current supplied by the photovoltaic source. Maintaining a positive balance between the current generated by the photovoltaic source and the current supplied to the load is essential for successful charging, especially under low-light-intensity conditions, reaching as low as 200 lux. Consequently, the photovoltaic harvester must deliver a current higher than the quiescent current $I_{q}$. This requirement leads to the selection of the amorphous silicon solar cell AM-1815 by Panasonic [42]. This cell offers a typical output current $I_{ope}$ of 45.7 μA at the voltage $V_{ope}$ of 3.0 V under a light intensity of 200 lux, with maximum overall dimensions of 58.1 mm × 48.6 mm × 1.1 mm. In addition, the A108G Ultra Low ESR COTS-Plus tantalum capacitor was selected for its high-performance characteristics, including a capacitance value of 1 mF and a voltage rating of 4 VDC, making it an ideal choice for $C_{storage}$ [43].

## 4. System Configuration Setup and Experimental Procedure

The primary function of the designed EAWSN is to establish communication with a base station, which enables the transmission of data in an energy-efficient and self-sustaining manner. Figure 4 illustrates the system components of the LoRaWAN protocol standard. In particular, Figure 4a shows the EAWSN platform, and Figure 4b shows the gateway used. The gateway consists of the NUCLEO-F746ZG board and the RisingHF LoRaWAN GS&HF1 (868/915/923 MHz) extension board with antenna from STMicroelectronics [44]. To ensure compatibility with the LoRaWAN protocol, the EAWSN must be configured to comply with protocol standards. This configuration increases the versatility and adaptability of the designed EAWSN platform for different applications.

### 4.1. EAWSN Configuration for LoRaWAN Protocol

The battery-free node was configured in CLASS A mode of the LoRaWAN protocol, as it is preferred for its energy-efficient characteristics, as described in [29]. Additionally, activation by personalization (ABP) was set to eliminate the need to create a session between the gateway and the EAWSN, and all receive windows of the EAWSN system were disabled for energy efficiency. Finally, the transmitted packet was configured to conform to the LoRaWAN packet format, as defined in Equation (Equation 7), which is 255 bytes in length for SF7, with a payload of 242 bytes and a header of 13 bytes.
(7)MHDR(1)+FHDR(7)+Port(1)+Payload(242)+MIC(4)

The transmit power level was set to 14 dBm, and adaptive data rate (ADR) was disabled to enable fixed spread factor transmission and avoid dynamic SF transmissions. In fact, the setting of transmission parameters such as spreading factor (SF) varied in the range of 7 to 12 between experiments. Finally, to improve data security through AES-256 encryption, a 16-byte application session key, network session key, and 4-byte device address were stored in the EAWSN’s nonvolatile memory.

### 4.2. Gateway Setup and Configuration

In our setup, the gateway, an NUCLEO-F746ZG board, is preconfigured to send data packets to the LORIOT network server by default. However, it is important to emphasize the adaptability of the system. It is possible to change the configuration of the gateway to enable compatibility with other network servers that utilize the Semtech packet forwarder protocol. For example, this may involve modifying the LoRaWAN server, MAC address, and gateway extended unique identifier (EUI) [45].

### 4.3. Network Server Setup

The server used is LORIOT EU1, located in Frankfurt, Germany. The configuration process includes entering the LoRaWAN server settings, MAC address, and gateway EUI into the base platform (packet forwarder STM). Finally, sensor devices can be registered by inputting device-specific parameters such as device address, network session key, and application session key, corresponding to those provided to EAWSN.

### 4.4. Maximum Achievable Packet Length

The initial phase involves the determination of the maximum reliable packet length (PL) that the EAWSN system can accommodate using LoRaWAN connectivity while simultaneously measuring energy consumption. Through the manipulation of PL and LoRaWAN parameters, our objective is to pinpoint the optimal configuration that strikes a balance between data throughput and power consumption. This preliminary phase is integral in gaining essential insights into the system’s data transmission capabilities under various conditions and will serve as a guiding factor for our battery-free sensor design in real-world implementations.

We started by setting up our equipment with some initial values: SF7, a PL = 10 bytes, and $C_{storage}$ = 2 mF. We used an oscilloscope to check the minimum voltage drop $V_{L}$. Then, the step-by-step process was initiated. At each step, PL increased by 10 bytes $V_{L}$, measured again until $V_{L}$ falls into the range of 1.8 to 2 volts to ensure the condition in Equation (Equation 5) when $V_{L}$ reaches this voltage range, the maximum PL that the EAWSN platform could handle with that particular $C_{storage}$.

To expand the scope of our research, the energy storage $C_{storage}$ increased by 2 mF, and the entire process is repeated. This incremental adjustment of $C_{storage}$ was carried out iteratively until we reached a final value of 16 mF. This stepwise approach was designed to investigate the impact of increased stored energy on the maximum attainable PL in our study. Furthermore, this experiment covered all SF ranging from SF7 to SF12. This approach allowed for a comprehensive examination of how variations in SF values and specific energy storage $C_{storage}$ influenced the maximum achievable PL. These efforts contributed significantly to a more comprehensive understanding of the performance of EAWSN with LoRaWAN connectivity under different conditions and configurations.

### 4.5. Determining the Coverage

In the second phase, the aim is to determine the maximum achievable communication range of the EAWSN, building on our findings from the first phase. We wanted to investigate the ability of the system to achieve extended communication distances while maintaining a constant configuration. This experiment is of paramount importance in understanding the operational limits of the EAWSN, with particular emphasis on its range performance. By investigating the maximum achievable communication distance, we will gain valuable insight into the system’s capabilities under varying conditions, thereby guiding our design choices for real-world implementations.

In this phase, the focus shifts to the evaluation of the EAWSN coverage in a specific environment an urban area in Catania, Italy, a noisy environment due to the presence of buildings and infrastructure. The configuration of the EAWSN includes fixed parameters: a BW of 125 kHz, a CR of 4/5, a transmit power (TP) of 14 dBm, and a PL of 20 bytes. The EAWSN will be maintained at a height of 15 m above ground.

The first test uses SF7 with an energy storage capacity ($C_{storage}$) of 2 mF. The next test will select SF8 and set $C_{storage}$ to 4 mF.

## 5. Experimental Results

In this section, experimental results regarding test coverage and the effect of storage capacitor dimensions on LoRaWAN packet length and spreading factor are presented.

### 5.1. Analyzing the Maximum Packet Length

In order to determine the maximum achievable packet length (PL) of the designed EAWSN platform, several parameters were kept constant during the measurement phase, including a bandwidth (BW) of 125 kHz, a transmit power (TP) of 14 dBm, and a code rate (CR) of 4/5. To determine the energy consumption $E_{LoRaTX}$ for each transmitted packet, a PicoScope oscilloscope was used to monitor the minimum voltage drop at each transmission packet. As already discussed in Section 3, the design must meet the conditions expressed in Equations (Equation 6) and (Equation 5). In particular, Equation (Equation 5) shows how the minimum voltage $V_{L}$ must always be above $V_{dd\_min}$, i.e., 1.7 V for the STM32WL55JC1. Figure 5 shows the experimental measurements of the voltage $V_{store}$ through an oscilloscope. This analysis explores various PL while keeping constant at 2 mF the $C_{storage}$ capacitance value. The radio is configured with spreading factor 7 (SF7), BW = 125 kHz, and TP = 14 dBm. The relationship between PL and voltage drop $V_{L}$ reveals an inverse proportionality so that while increasing the packet length, the voltage level $V_{L}$ decreases. As shown in Figure 5a,b when the radio transmits data with a packet length of 10 bytes (PL = 10 bytes), the voltage $V_{L}$ decreases to ≈2.5 V, while for a higher packet length of 30 bytes, the voltage drop diminishes further to the value $V_{L}\approx$ 2 V. Based on this observation, it is essential to avoid using a packet length greater than 30 bytes to prevent the system from entering reset mode.

Figure 6a and Figure 6b show, respectively, the measurement results of the voltage $V_{L}$ and the corresponding energy consumption during the transmission phase by varying PL. These experimental results were obtained using a 4 mF storage capacitor ($C_{storage}$). Doubling the storage capacitance from 2 mF to 4 mF results in a doubling of the stored energy, enabling an increase in packet length from 30 to 80 bytes.

Figure 7 shows the experimental results of measuring packet lengths corresponding to $C_{storage}$ values and their associated spreading factors (SF7 to SF12). The graph clearly shows that with a $C_{storage}$ of 4 mF, an SF of 7 allows a maximum PL of 80 bytes. This maximum PL is reduced to 30 bytes at SF8 and further reduced to only 10 bytes at SF9. Beyond SF9, the system shows an inability to transmit data packets of any length. This limitation is due to the modulation scheme associated; when SF increases, the time-on-air (TOA) also increases, resulting in higher power consumption.

These measurements provide valuable insights into the behavior of the designed EAWSN platform. Increasing $C_{storage}$ extends the maximum PL while keeping SF constant. On the other hand, maintaining $C_{storage}$ at the same level but increasing SF approximately halves the required packet length. These results have important implications for optimizing the performance of our system.

### 5.2. Analyzing the Coverage

To evaluate the communication range of our EAWSN system, we carried out a series of tests in different locations in a busy urban environment in the central area of Catania. Table 3 shows a conductive test carried out in the scenario of Figure 8 with the following specific parameters: SF7, a bandwidth of 125 kHz, a transmission (TX) output power of 14 dBm, CR = 4/5, PL = 20 bytes and $C_{storage}$ = 2 mF. The GW is positioned at a height (h) = 15 m. The test provided detailed data on RSSI and SNR for each tested point (points 1 to 4). The data presented in the table indicate that at point 4 at a distance of ≈560 m, the minimum achievable SNR and RSSI were −8 dB and −110 dBm, respectively.

Table 4 shows the test carried out in the urban scenario of Figure 9 with the following specific parameters: SF8, a BW of 125 kHz, a TP of 14 dBm, CR = 4/5, PL = 20 bytes and $C_{storage}$ = 4 mF. The test provided detailed data on RSSI and SNR for each tested point (points 1 to 5). The data indicate that at point 5 at a distance ≈1100 m, the minimum achievable SNR and RSSI were −13 dB and −113 dBm, respectively.

The results obtained are in line with the expected performance for LoRaWAN devices and demonstrate the practical performance of our EAWSN system in an urban environment. These results demonstrate the applicability of our system design in real-world scenarios, especially in challenging urban environments. Furthermore, we calculated the linear correlation coefficients (*r*) between RSSI and SNR for the two experimental scenarios SF7 and SF8, which were 0.95 and 0.94, respectively. These high correlations indicate consistent and coherent signal quality in both test scenarios. Based on the provided SNR and RSSI data, we have generated informative graphs representing the relationship between SNR and RSSI regarding distance. This inverse correlation is observed in Figure 10a,b. These results are consistent with the basic principles of LoRa technology and modulation techniques. They show that increasing the spread factor results in an extended communication range but at the cost of higher power consumption. These trade-offs highlight our EAWSN system’s adaptability in different environmental conditions and provide valuable insights into its performance characteristics in the complex urban context we studied.

## 6. Discussion

This section addresses the optimal configurations for the EAWSN under different lighting conditions, discusses potential constraints, and evaluates its scalability.

### 6.1. EAWSN under Different Light Condition and Recommended Setting

We tested the EAWSN system both in indoor environments, where the illuminance was approximately 500 lux (similar to a bright office), and in outdoor environments, where sunlight typically ranges from 32,000 lux to 100,000 lux. However, the EAWSN can also operate under low-light-intensity conditions, around 200 lux, as the solar cell can provide an operating voltage of 3 V and an operating current of 45.7 µA [42]. These tests demonstrate that the EAWSN system can maintain an acceptable level of energy autonomy in diverse environmental conditions, ensuring functionality even in low-light scenarios. Moreover, in our experiments, we evaluated various transmission configurations to determine the most suitable one for the needs of the EAWSN. Our study can help identify appropriate design parameters such as capacity and maximum package size. Along with these parameters, design optimization should also consider other factors, such as cost. However, the recommended design must take into account the use case. For example, the packet size can vary from a few bytes in smart agriculture [46,47] to 200 or more bytes in industrial contexts [48]. Therefore, a recommended choice should find a good compromise between maximum payload and device coverage radius. If flexibility on packet size, coverage, and spreading factor allocation is chosen, a 16 mF capacitor could be the recommended choice. With this configuration, the packet size ranged from a minimum of 10 bytes to a maximum of 255 bytes for SF12 and SF7, respectively. We identified that optimizing transmission strategies is crucial to ensure efficient operation of the system in autonomous mode. Conversely, if the goal is to reduce size and costs, a good compromise between coverage and maximum payload length could be achieved using an 8 mF capacitor with an SF8 setting. In this case, a packet size of 90 bytes and a maximum range of 1100 m in urban environments can be achieved.

### 6.2. Limitations and Possible Solutions

The possible limitations of the proposed system are basically twofold: the packet size is limited to a few bytes for higher spreading factors (e.g., 10 bytes at SF12), and the system does not operate in the absence of ambient light, such as at night. However, there are solutions for these two limitations. Both can be addressed by choosing a capacitor with a higher capacitance value, allowing for a larger packet size even at high SF values, according to the LoRa regional parameters policy in [49]. The size of the capacitor can be selected according to the specific use case, as discussed in Section 6.1. Furthermore, by integrating a supercapacitor into the system and jointly using adaptive transmission algorithms, as proposed in [6], it is possible to enable data transmission even during the night.

### 6.3. Mass-Scale Capability Eliminating Battery Replacement

The widespread adoption of IoT sensors has been favored by the almost ubiquitous availability of wireless connectivity and the rapid decline in sensor costs [50,51]. It is predicted that by 2025, there will be 75 billion IoT devices worldwide [52]. However, there are limitations to the massive deployment of IoT sensors. In fact, battery-powered sensors significantly increase the maintenance costs of IoT devices and reduce their lifespan [50,53]. The cost of replacing batteries is often higher than the cost of the IoT device itself, which limits the large-scale deployment of IoT devices. According to forecasts, there will be about 274 million battery replacements in IoT devices per day in a 10-year lifespan scenario, and the number would rise to 913 million per day in a 3-year lifespan scenario [50]. In this context, our work shows how the adoption of energy harvesting, together with ultra-low-power microcontroller systems, can eliminate the need for batteries, promoting a large-scale and environmentally friendly deployment of IoT sensors in the near future.

## 7. Conclusions

A self-powered wireless sensor node (WSN) was designed, built, and tested. This device exemplifies the EAWSN platform’s capability to transmit data using LoRaWAN technology without relying on batteries or external power sources. The conducted experiments shed light on the significance of optimizing LoRaWAN transmission settings, including spreading factor (SF), bandwidth (BW), coding rate (CR), packet length (PL), and transmit power (TP), in managing energy consumption. The proposed solution lays the foundation for widespread deployment in mass-scale IoT scenarios.

The experimental trials conducted in Catania, Italy, showcased the exceptional performance of the EAWSN. Specifically, with carefully configured settings such as a transmit power of 14 dBm, SF7, 125 kHz BW, CR of 4/5, a PL of 20 bytes, and a 2 mF energy storage capacitor, the system achieved a communication range of 560 m. Furthermore, adopting a similar configuration with SF8 and a 4 mF energy storage capacitor, the system impressively extended its range to 1110 m. These achievements, coupled with the remarkable communication range, underscore the system’s robustness and versatility, rendering it suitable for a multitude of applications. Additionally, this research highlights the EAWSN’s adaptability to function effectively in two distinct environments. One environment entails direct communication between WSNs, while the other operates within the LoRaWAN protocol framework, each with its specific constraints. This showcases the system’s versatility across various settings, making it a flexible choice for diverse scenarios.

In conclusion, the designed system represents a clean technology solution that not only offers sustainability and efficiency but also aligns with contemporary environmental concerns. By eliminating the need for batteries and offering maintenance-free operation, it presents an eco-friendly approach that is ideal for responsible and practical deployment across a wide spectrum of applications. Moreover, this sensor has the potential to revolutionize the field of mass-scale IoT sensing and enable new applications that were previously impossible due to power constraints.

In the future, research will continue, with a focus on specific areas of investigation. One such aspect is the potential implementation of this system using the LoRaWAN standard with OTAA authentication, which provides enhanced data security. In addition, research will continue into adapting the system to operate in low- or no-light conditions.

## Figures and Tables

**Figure 1 sensors-24-04279-f001:**
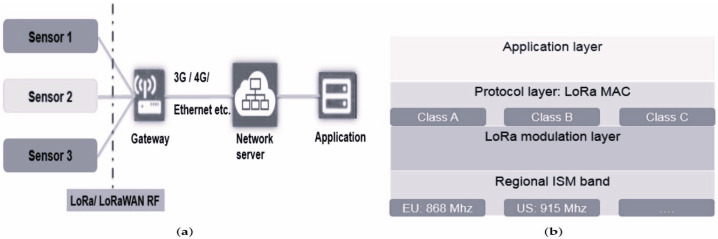
(**a**) LoRaWAN network architecture. (**b**) LoRaWAN protocol stack.

**Figure 2 sensors-24-04279-f002:**
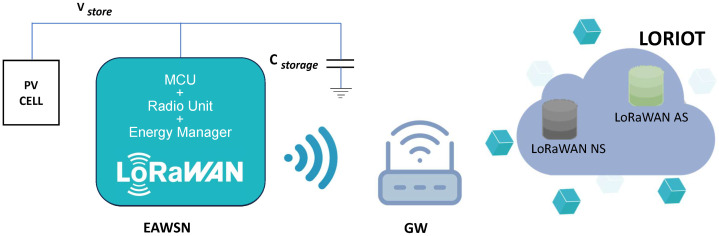
LoRaWAN architecture with block diagram of EAWSN, gateway (GW), and LORIOT cloud, which includes network server (NS) and application server (AS).

**Figure 3 sensors-24-04279-f003:**
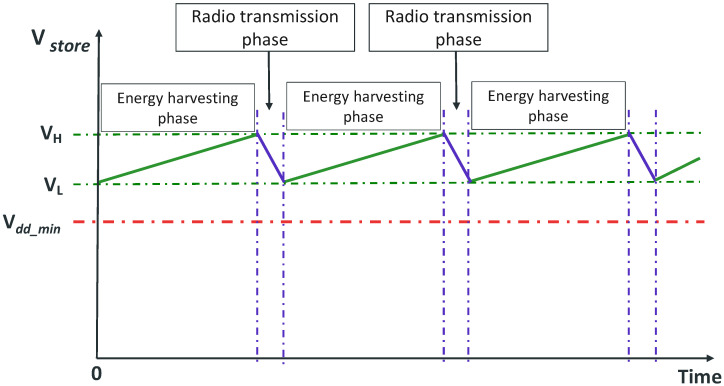
Functional description of the designed EAWSN platform.

**Figure 4 sensors-24-04279-f004:**
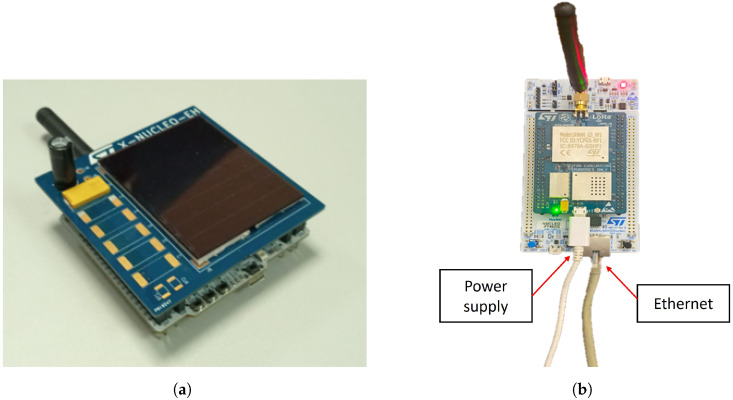
System components for LoRaWAN protocol configuration. (**a**) EAWSN platform with (NUCLEO-WL55JC) board, amorphous silicon solar cell (AM1815), and $C_{storage}$ ranging from 2 mF to 16 mF. (**b**) Gateway (NUCLEO-F746ZG) as receiver.

**Figure 5 sensors-24-04279-f005:**
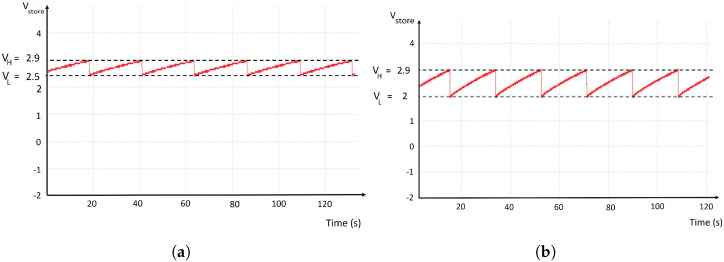
Experimental measurements with the oscilloscope of the voltage $V_{store}$. Setup conditions: SF = 7, BW = 125 kHz, TP = 14 dBm, $C_{storage}$ = 2 mF. Figure (**a**) shows the voltage $V_{store}$ for PL = 10 bytes, $V_{H}\;\approx \;3\;V$. $V_{L}\;=\;2.5\;V$. Figure (**b**) shows the voltage $V_{store}$ for PL = 30 bytes, $V_{H}\;\approx \;3\;V$. $V_{L}\;=\;2.0\;V$.

**Figure 6 sensors-24-04279-f006:**
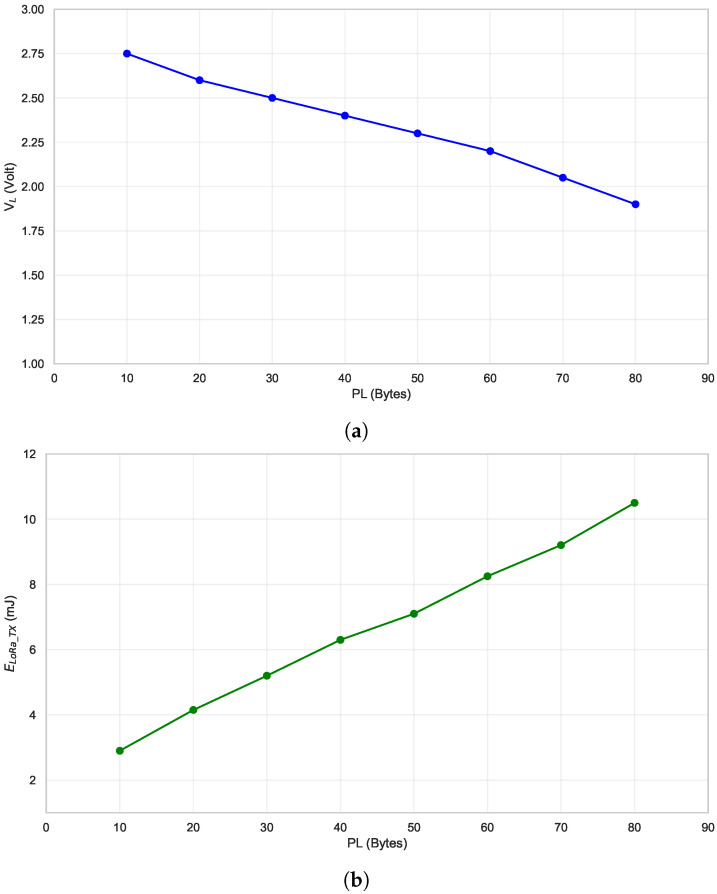
Measurement results: Variations in $V_{store}$, $V_{L}$, and $E_{LoRa_{T}X}$ vs. packet length (PL). Setup conditions: SF = 7, BW = 125 kHz, TP = 14 dBm, $C_{storage}$ = 4 mF. Figure (**a**) shows minimum voltage $V_{L}$ vs. PL. Figure (**b**) shows energy required $E_{LoRa\_TX}$ for transmission vs. PL.

**Figure 7 sensors-24-04279-f007:**
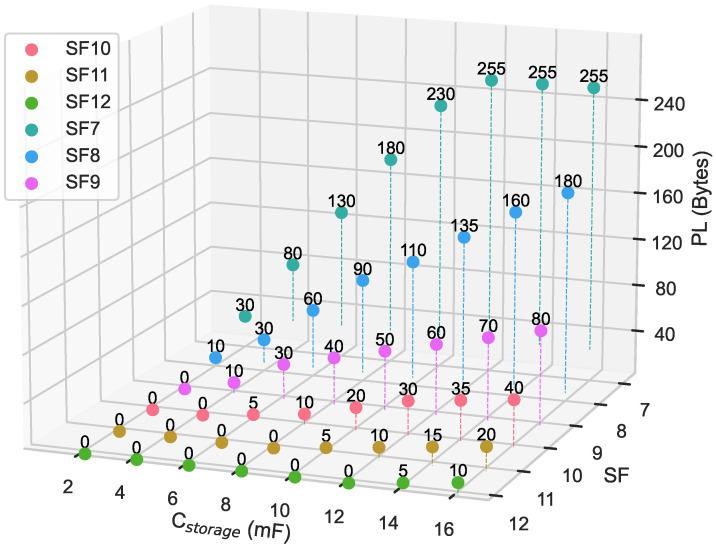
LoRaWAN maximum PL as function of $C_{storage}$ for SF from 7 to 12. Setup conditions: BW = 125 kHz, TX output power = 14 dBm, CR = 4/5.

**Figure 8 sensors-24-04279-f008:**
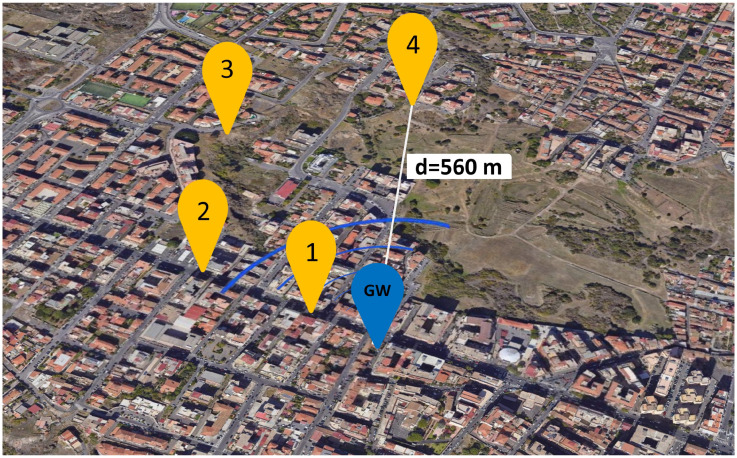
Communication distance in urban area for SF7, BW = 125 kHz, $C_{storage}$ = 2 mF, TX output power = 14 dBm, CR = 4/5, PL = 20 bytes, GW at h = 15 m.

**Figure 9 sensors-24-04279-f009:**
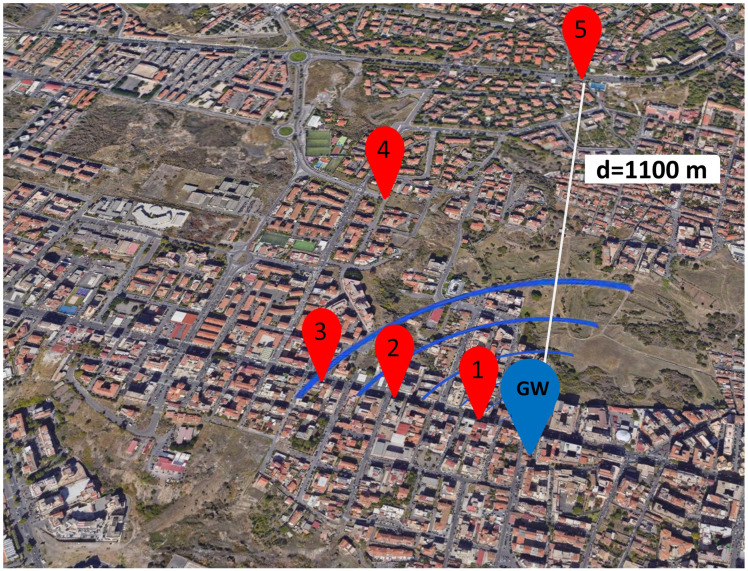
Communication distance in an urban area for SF8. Setup conditions: spreading factor SF = 8, BW = 125 kHz, $C_{storage}$ = 4 mF, TX output power = 14 dBm, CR = 4/5, PL = 20 bytes, GW at h = 15 m.

**Figure 10 sensors-24-04279-f010:**
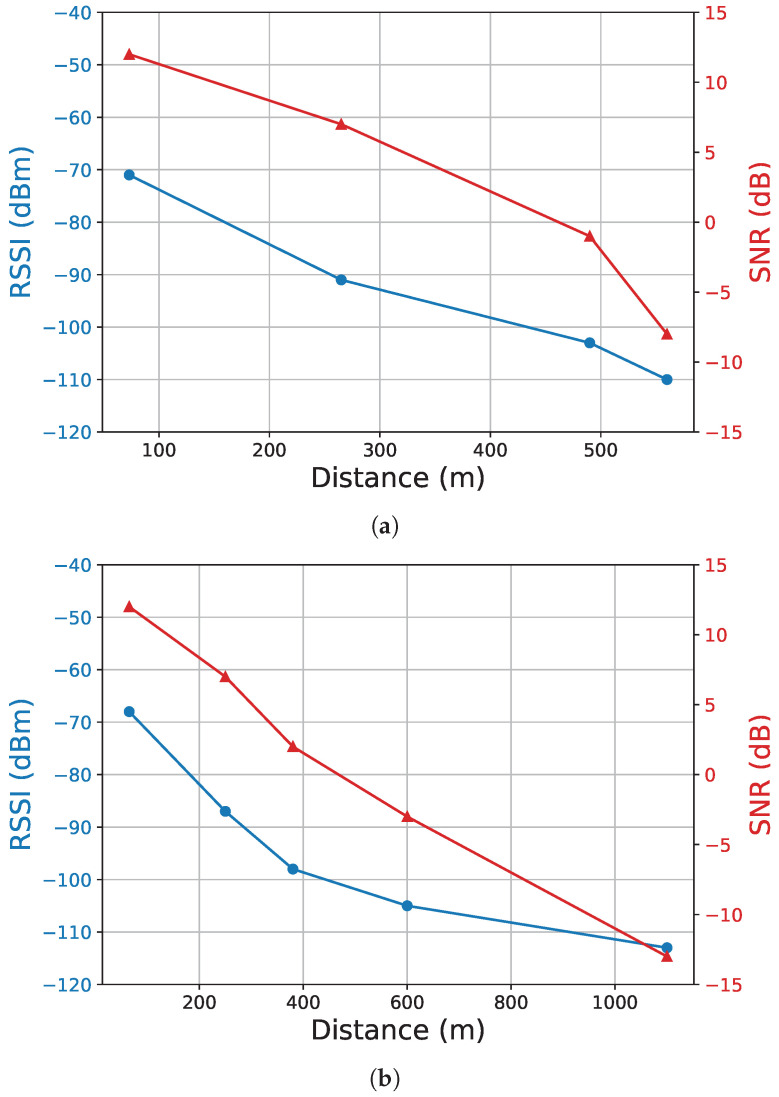
SNR and RSSI values at varied distances. Setup conditions: BW = 125 kHz, TP = 14 dBm, CR = 4/5, PL = 20 bytes, TX at h = 15 m. Panel (**a**) depicts SNR and RSSI values vs. distance with SF = 7. Panel (**b**) depicts SNR and RSSI values vs. distance with SF = 8.

**Table 1 sensors-24-04279-t001:** LoRaWAN Classes Comparison.

Class	Description	Energy Consumption
A	Sensor triggers, Followed by a downlink response.	Most efficient
B	Communication occurs in slots Simple-synchronized beacon	Controlled downlink
C	Communication without delay.ensuring Downlink communication without delay.	High power consumption

**Table 2 sensors-24-04279-t002:** Spreading Factors and corresponding chips length.

Spreading Factor (SF)	Chips Length $2^{\mathrm{SF}}$
7	128
8	256
9	512
10	1024
11	2048
12	4096

**Table 3 sensors-24-04279-t003:** Experimental results of coverage, SNR, and RSSI for various urban points in the Figure 8 scenario (SF7, $C_{storage}$ = 2 mF, PL = 20 bytes).

Point	Distance (m)	SNR (dB)	RSSI (dBm)
1	73	12	−71
2	265	7	−91
3	490	−1	−103
4	560	−8	−110

**Table 4 sensors-24-04279-t004:** Experimental results of coverage, SNR, and RSSI for various urban points in the Figure 9 scenario (SF8, $C_{storage}$ = 4 mF, PL = 20 bytes).

Point	Distance (m)	SNR (dB)	RSSI (dBm)
1	65	12	−68
2	240	7	−87
3	380	2	−98
4	600	−3	−105
5	1100	−13	−113

## Data Availability

Data are contained within the article.

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
