# Peer review of "Towards Mass-Scale IoT with Energy-Autonomous LoRaWAN Sensor Nodes"

_sensors, 2024, doi:10.3390/s24134279_

Round 1

Reviewer 1 Report

Comments and Suggestions for Authors

This paper proposes an energy-autonomous wireless sensor system design and validated the system performance through validation in an urban environment. The paper is generally well-presented; however, the reviewer has the following suggestions for improvement.

1.       Has the authors assessed the life expectancy of the EAWSN under different light condition? Will the system always be energy-autonomous under different environments?

2.       What is the recommended setting for the proposed EAWSN to operate energy-autonomously, such as for transmission?

3.       Please discuss the limitations of the proposed system.

Author Response

A point-by-point response in attached file 

Reviewer 2 Report

Comments and Suggestions for Authors

The paper tackles a very relevant topic. However, there are still some open questions reading this paper. 

Minor:

- many minor typos, inconsistancies in capital or lower case letters, abbreviations etc. I suggest to do a comprehensive language check.

- structure intro: you reference to your approach (line 56ff) but no approach etc. is clear at this point

- quality and explanation of figures can be improved. Some figures are not explained properly or general quality is low. 

- you say spreading factors are orthogonal but there are works talking about quasi-orthogonal. You should at least mention/discuss that in my opinion

- Some information are not reference (e.g. only 5% of batteries are recycled)

- there is literature regarding LoRaWAN transmission and power consumption using different sensors and channel access approach from different groups that are not listed in your literature section. Are you aware of these works? You should maybe consider adding (at least parts of them) to your related work (e.g., Polonelli et al, Ortin et al, Loh et al, Leonardi et al, Pham et al)

- line 222 you talk about use cases - but they are not clear to me at this point. I suggest to improve the red line in your work. 

Major concerns:

- in general, it is hard to follow and to understand what you do, why your results are good and, in particular, better than results from literature. Starting with the introduction, a red line is missing. I suggest structuring the last part of the introduction as follows: First, detail what you do and then, clearly define your contribution. Currently, you first talk about 'our innovative approach take into account...' but the approach is not clear at this point. Finally, clearly define the research questions you tackle and answer in your work. That is not clear to me.

- To the best of my knowledge, a higher spreading factor does not lead to a higher signal to noise ratio but to the antenna being aware of decoding signals arriving with a higher signal to noise ratio

- This is my main concern with your work, starting at section 3: from there, it is rather hard to follow, missing red line, occasionally typos and methodology is not clear. Here are the issues I would like to raise:

- You alternate between harvesting and transmitting phase. Then you say: when you harvested energy you transmit. But if I do not have to transmit at that point. Or if I have to transmit at any point earlier that are both valid use cases using LoRaWAN. I do not understand how you can cope with such situations.

- What happens if - for any reason, be it that the cheap sensor is doing strange stuff that is totally reasonable to the best of my knowledge - you fall below your V_ddmin. Does the system break then? 

- Your energy model reads like: you harvest, then you have a specific amount of energy available and can transmit. Then you need to harvest again. And the capacity stored defines how large your packet can be. I do not understand the novelty of this statement - that is rather straight forward to me.

- You talk about varying payload, SF, and other LoRaWAN message parameters. There are studies for this and the impact on the time on air and consequently, dependent on the hardware, on the energy consumption. I do not get the delta to these works (see one of the works from the minor comments authors list)

- In general, all your considerations are highly hardware dependent. How does your system behave using different hardware (there is also a work in the literature discussing this IEEE IoT Journal, Loh et al.)

- line 267,268, you talk about LoRaWAN protocol standard. I do not get this point.

- You say that you vary your payload up to 255 bytes. That is wrong from a methodology perspective. SF12 only allows messages up to (I think) 52B. But you do not detail on that. 

- I do not understand why the information about encryption is relevant for your work. So in general I do not understand Section 4.1

- Why is it important that you transmit your data to a server afterwards. You just study whether the sensor can be autonomous, right? I do not understand why this information helps the reader.

- line 305: You talk about energy consumption to measure. Do you also consider this? Does that impact your results? So it is hardware dependent again, right? I thought, you just consider transmissions (as mentioned before)

- In Section 4.4, it is not clear, why your setup is as it is (why you use these parameters/start with these parameters and not with others?) Do other parameters behave differently? 

- I doubt your coverage quantification scenario is valid. The transmission distance is dependent on so many factors (and you did not even specify the sensor height that is one of the most relevant ones - see gateway placement and coverage approaches in the literature for LoRaWAN, e.g., also by some of the authors in the literature list above.). I know from measurements where just the presence of persons or busses in the line of sight changed results drastically. Furthermore, it is dependent on geography, urbanization, weather, antenna quality and gain, and so on. To claim these points, you should do large scale studies and a valid statistical analysis (details follow in one of my next comments)

- Section 5, figure 5, you state that you analyze the payload. But there is no payload shown in the figure

- You say that your system does only work with SF7-9. What do I do if I want to transmit with larger SFs?

- Are your results in figure 6 deterministic? If yes, can you derive a general equation from it. If not, how much variance is in your result. Are differences statistically significant? Can you just 'connect' the points in your figure?

- line 395/396: You say that your results match theoretical expectations. I am not aware of any works that state that. They just state that e.g., 'larger SF = longer transmission' but that is no novel result and already shown in many works. In the same line, you talk about a validation. I do not understand which part has been validated. 

- You talk about usage for 'mass-scale IoT' but only conduct studies with 5 points. Can you please explain your study design that is used to achieve the results for figure 10. How many messages did you transmit? How does other factors impact the results (geography etc.). Did you do any statistical analysis? Are results statistically significant? How can you conclude that it is valid for 'mass-scale IoT'?

Comments on the Quality of English Language

Comments see above.

Author Response

(The authors gave the same response as above.)

Reviewer 3 Report

Comments and Suggestions for Authors

- Authors propose an Energy-Autonomous Wireless Sensor Node (EAWSN), suitable for long-term mass-scale IoT applications in remote and inaccessible environments.

- It uses LoRaWAN protocol, which nodes are powered by photovoltaic cells, harvesting ambient light in an indoor environment.

- Authors perform real-life experiments.

- Authors did not cite the Section 2 in the structure presented in the last

paragraph of Introduction.

- Authors should review the text, in order to correct some typos.

- Authors should verify some equivalent phrases in Section 2.

- Authors must check the first phrase of the Subsection 5.1

- How many transmissions were performed for the experiments, and during how much time?

- The table presented along with Figures 8 and 9 could be showed in separate, as a table environment.

- We can not see the caption a) in Figure 10.

- Authors could calculate the correlation between RSSI and SNR, for the results presented in Figure 10.

- Authors should present a better discussion regarding to the mass-scalecapability of their proposal.

Comments on the Quality of English Language

The paper is well written, with a very good quality of English language.

Author Response

(The authors gave the same response as above.)

Round 2

Reviewer 2 Report

Comments and Suggestions for Authors

No further comments. All comments of my first review have been tackled. Thank you to the authors.

Reviewer 3 Report

Comments and Suggestions for Authors

I am very satisfied with this version of the manuscript.